# REVISITING ONLINE AND OFFLINE CONTINUAL LEARNING THROUGH ALIGNED RESOURCES

## ABSTRACT

Instead of training deep neural networks offline with a large static dataset, continual learning (CL) considers a new learning paradigm, which continually trains the deep networks from a non-stationary data stream on the fly. Despite the recent progress, continual learning remains an open challenge. Many CL techniques still require offline training of large batches of data chunks (i.e., tasks) over multiple epochs. Conventional wisdom holds that online continual learning, which assumes single-pass data, is strictly harder than offline continual learning, due to the combined challenges of catastrophic forgetting and underfitting within a single training epoch. Here, we challenge this assumption by empirically demonstrating that online CL can match or exceed the performance of its offline counterpart given equivalent memory and computational resources. This finding is further verified across different CL approaches and benchmarks. Conceptually, we demonstrate that online and offline CL follow the same underlying algorithmic framework when provided with equivalent memory and computational budgets. The sole difference lies in the space allocation hyperparameter $\alpha = M_{short}/M$, which controls the amount of space reserved for recent samples. Theoretically, we show that a smaller $\alpha$ yields a lower generalization bound, explaining the superior performance of online CL.

## 1 INTRODUCTION

Deep neural networks have achieved remarkable performance in many applications, yet learn very differently than biological brains. Humans acquire knowledge continually from varied experiences, gradually mastering new skills over a lifetime without forgetting old ones. In contrast, deep learning algorithms are typically trained via a static paradigm - given a fixed dataset, models are trained for multiple epochs over shuffled datasets to converge. This learning paradigm assumes the availability of all training data upfront and its independent and identically distributed (IID) nature. However, in many real-world applications, data arrives continuously as an infinite data stream and is not static. Incorporating new information from such streams into neural networks is often a destructive process that catastrophically interferes with or erases previously learned representations, known as catastrophic forgetting (French, 1999; Delange et al., 2021). One way to avoid catastrophic forgetting is to naively retrain the model from scratch on cumulative data whenever new data arrives, which can result in unsustainable storage and computation costs. The field of continual learning (CL), also known as lifelong learning, aims to overcome this challenge by developing specialized algorithms and architectures for non-stationary environments with limited memory and computation resources.

The general goal of continual learning is to learn efficiently from an infinite data stream, where the data distribution may change gradually or abruptly at any time. Due to the challenge of non-stationary data streams, several practical CL settings have been designed with simplifying assumptions. One common assumption is that the data stream consists of sequential tasks, with each period of consistent data distribution considered as a task. Many techniques developed in these settings require multiple passes or epochs over the task data (Delange et al., 2021; Masana et al., 2022) assuming that the task boundaries are known. This is referred to as *offline continual learning*. While these methods have shown promising results in mitigating forgetting, assuming clear task divisions and the knowledge of task switches limits their applicability to real-world continual learning. Another, more general, setting is *online continual learning*, also known as task-free continual learning,

which operates fully incrementally on the fly as data streams in. Each sample or mini-batch is observed only once, precluding offline training. A typical online CL method is Experience Replay (ER) (Mai et al., 2022; Chaudhry et al., 2019). In ER, an incoming batch from the data stream is concatenated with a batch of exemplars for gradient updates. This setting aligns well with practical continual learning desiderata (Delange et al., 2021) and has gained increasing attention. However, one common belief is that online continual learning is more challenging than offline CL (refer to Mai et al. (2022) for an empirical survey for online CL and Masana et al. (2022) for offline CL). The combined challenges of catastrophic forgetting and the inability to fully adapt to each task in one pass are believed to limit the performance of online methods. As mentioned in Buzzega et al. (2020), *"Despite being interested in an online scenario, with no additional passages on the data, we reckon it is necessary to set the number of epochs per task in relation to the dataset complexity."*.

Upon reviewing the current practices in the CL community, we find that offline CL generally demands higher memory and computational resources compared to online CL. Specifically, offline CL algorithms typically use an epoch number of 70-200, while most online CL methods only use a single iteration for each incoming batch ($I = 1$), and some recent works (Zhang et al., 2022; Soutif-Cormerais et al., 2023) consider $I = 3, 9, 10$. In terms of memory usage, both online and offline CL works mainly focus on the cost of exemplar samples. However, apart from historical exemplars, new task samples also exist within the memory-constrained operational space. Offline CL necessitates large additional space to store the task samples, whereas online CL only requires a small space for new incoming batch.

Under the aligned memory and computation framework, we systematically compare online and offline CL across various data stream sizes, task sizes, datasets, and CL algorithms. Interestingly, we find applying the same CL methods (e.g. ER, SCR, iCaRL, DER++) in an online manner consistently leads to better performance than in an offline manner.

From a conceptual perspective, we revisit online and offline CL with an aligned memory budget. Fundamentally, they both maintain a First-In-First-Out (FIFO) buffer to store recent samples while using another mechanism (e.g. reservoir sampling) to store a subset of previous non-recent samples. Specifically, online CL uses a FIFO buffer of size equal to the batch size $B$ to store recent samples. Offline CL uses a buffer of size equal to the task size $C$. This reveals an underlying connection - online and offline CL follow the same framework but differ in how much space $\alpha = M_{\text{short}}/M$ they allocate to store recent samples. Intuitively, the choice of $\alpha$ relates to the stability-plasticity tradeoff in CL: higher $\alpha$ enhances plasticity but reduces stability. We investigate the effectiveness of semi-online cases with $B < M_{\text{short}} < C$ that are currently unexplored in the literature. We find, perhaps surprisingly, that pure online CL still outperforms these semi-online choices.

To better understand this phenomenon, we derive a generalization bound for CL based on discrepancy distance and total memory budget. This bound suggests better generalization for smaller $\alpha$, explaining the superior performance of online CL. Additionally, we conduct further analysis and experiments that confirm the benefits of a small $\alpha$ persist across various continual learning scenarios. Specifically, we find that online CL provides increasing gains over offline CL given longer input streams, smaller memory budgets, and larger task sizes. We hope that the unexpected finding presented in this paper will raise awareness in the CL community, encouraging the development and evaluation of CL techniques in online task-free settings, and pushing the field towards general continual learning.

Our contributions are as follows. First, we compare online and offline continual learning under aligned memory and computation budgets and empirically demonstrate the effectiveness of the former. Second, we introduce a unified framework showing online and offline CL are the variants of the same algorithm only differ in storage allocation parameters. Third, we highlight the importance of storage allocation parameters on CL performance. Four, we provide a theoretical study on the generalization bound of online and offline CL, corroborating our experiment findings.

## 2 RELATED WORK

**Continual learning settings**. General continual learning (Delange et al., 2021; Buzzega et al., 2020) is an ideal scheme for learning from an infinite data stream, with desiderata like constant memory, online learning, no task boundaries, no task labels, and graceful forgetting. Various relaxations exist

with different assumptions. Early work focused on task-incremental settings (Mallya & Lazebnik, 2018; Serra et al., 2018) that assume access to task labels during training/testing. Despite promising results, relying on a task oracle is impractical. Recent class-incremental and domain-incremental learning approaches remove this assumption and try to learn cross-task knowledge and mitigate forgetting (Mirza et al., 2022; Masana et al., 2022; Van de Ven & Tolias, 2019). Nevertheless, these methods still require the knowledge of task boundaries to allow multi-epoch training over tasks. Online continual learning paradigm (Chaudhry et al., 2019; Aljundi et al., 2019; Mai et al., 2022) eliminates task boundary assumptions, performing single-pass learning over streams.

**Continual learning techniques**. Continual learning algorithms address catastrophic forgetting in three main ways: Rehearsal-based methods (Chaudhry et al., 2019; Aljundi et al., 2019) store and replay past samples to mitigate forgetting; Regularization-based methods (Rebuffi et al., 2017; Li & Hoiem, 2017) use regularization losses to encourage retention of past knowledge; Architecture-based methods (Mallya & Lazebnik, 2018; Serra et al., 2018) separate parameters for different tasks to avoid interference.

**Theoretical study in continual learning**. There are several theoretical studies in continual learning. Some works consider the linear approximation of the neural network around its initialisation and formulate CL as a recursive Kernel Regression. Under this framework, Doan et al. (2021) theoretically compared SGD and orthogonal gradient descent and Karakida & Akaho (2021) analyzed learning curves. Several other works (Peng et al., 2023; Doan et al., 2021) provide more general analysis with PAC-Bayes bounds but primarily focus on offline sequential tasks. Pentina & Lampert (2014) proposes a PAC-Bayesian framework to provide a learning bound on expected error in future tasks. Peng et al. (2023) constructs an ideal continual learner framework and derives generalization bounds for rehearsal in offline CL setting. The most similar work to ours is Ye & Bors (2022), which also derives a generalization bound based on discrepancy distance and Rademacher Complexity. However, Ye & Bors (2022) focuses solely on online CL and proposed to reduce discrepancy via parameter isolation. Our analysis unifies online and offline CL and reveals the influence of memory allocation on the generalization capability in continual learning.

## 3 A COMPARATIVE STUDY OF ONLINE AND OFFLINE CONTINUAL LEARNING

### 3.1 PROBLEM SETTING

Online and offline continual learning have been studied as separate research areas. We formalize the problem settings and terminology to describe the two paradigms as follows.

**Definition 1** (General Continual learning). Given a non-stationary (potentially infinite) stream of data $\mathcal{D}_t = \cup_t \mathcal{X}_t$: at each time step $t$, a continual learning algorithm $\mathcal{A}$ receives an incoming batch of data samples $\mathcal{X}_t = \{\mathbf{x}_i, y_i\}_{i=1,...,|\mathcal{X}_t|}$ that are drawn from the current data distribution $\mathbb{P}_t(x, y)$. The goal is to minimize the empirical risk on all the data seen so far:

$$\min_\theta \mathcal{R}(\theta) = \min_\theta \frac{1}{\sum_t |\mathcal{X}_t|} \sum_t \sum_{\mathbf{x},y \in \mathcal{X}_t} L\left(f(\mathbf{x}; \theta), y\right) \doteq \min_\theta L(\cup_t \mathcal{X}_t; \theta). \quad (1)$$

with a loss function $L$, a CL network function $f : x \to y$, and its associated parameters $\theta$.

**Definition 2** (Sequential tasks). Consider a sequential partition $\mathcal{G}$ of data steams $\mathcal{D}_t$ which satisfies $\cup_{C \in \mathcal{G}} C = \mathcal{D}_t$ and $C_i \cap C_j = \emptyset$ for any $C_i, C_j \in \mathcal{G}$ with $i \neq j$. The boundary of consecutive task $C_i$ and $C_{i+1}$ is denoted by the timestamp $T_i \in [0, t]$, and $T_i > T_j$, for any $i > i$.

While online continual learning deals with data streams $\mathcal{D}_t = \cup_t \mathcal{X}_t$ directly to solve the general continual learning problem in Definition 1, offline continual learning deals with a set of tasks and aims to maximize performance on all the tasks seen. Formally, we defined the transformation from data streams $\mathcal{D}_t$ to sequential tasks $C_i$ in Definition 2. Each task data can be denoted as $C_i = \cup_{t \in [T_i, T_{i+1}]} \mathcal{X}_t$. The data distribution within a task is assumed to be static and is denoted by $\mathbb{P}_{C_i}$.

**Definition 3** (Exemplars). Given a bounded exemplar space $M$, a stream of data $\mathcal{D}_t = \cup_t \mathcal{X}_t$ ($|M| < |D|$), and sample selection policy $\pi$: at each time step $t$, an online exemplar management algorithm takes in parts of the incoming batch into the memory and ejects some of the previous data $M_t \subset_\pi M_{t-1} \cup \mathcal{X}_t$; at each task $i$, an offline exemplar management algorithm takes in parts of the task data into the memory and ejects some of the previous data $M_i \subset_\pi M_{i-1} \cup C_i$.

When the exemplar space is sufficiently large to store all past data (i.e., $|M| > |D|$), the non-stationary data issue can be solved by storing all the data and performing offline training over shuffled batches. This is often regarded as the upper bound of continual learning performance. Therefore, continual learning considers the situation of infinite data streams with a bounded memory budget (i.e. $|M| \ll |D|$). Only a subset of data stream samples can be selected to be memorized. The stored exemplars can be trained alongside the incoming samples to alleviate forgetting, as shown in Eq 3 and Eq 2. This is referred to as rehearsal or experience replay. A simple yet effective sample selection strategy is reservoir sampling (Vitter, 1985), which randomly selects a sample of $k$ items from a larger population of unknown or very large size. Some CL works consider the case of $\mathcal{M} = \emptyset$ and only perform training on the incoming data, termed rehearsal-free CL.

**Definition 4** (Offline continual learning). Given a data stream with task boundary $\mathcal{D}_t = \cup_i C_i$, an exemplar set $\mathcal{M}$, and a loss function $L_\theta$, the gradient-based update rule in offline continual learning is defined as:

$$\theta \leftarrow \theta - \eta \nabla L(\mathcal{B}_{C_i} \cup \mathcal{B}_M; \theta), \text{where } \mathcal{B}_{C_i} \sim C_i, \mathcal{B}_M \sim \mathcal{M}. \tag{2}$$

**Definition 5** (Online continual learning). Given a data stream $\mathcal{D}_t = \cup_t \mathcal{X}_t$, an exemplar set $\mathcal{M}$ and a loss function $L_\theta$, the gradient-based update rule in online continual learning is defined as:

$$\theta \leftarrow \theta - \eta \nabla L(\mathcal{X}_t \cup \mathcal{B}_M; \theta), \text{ where } \mathcal{B}_M \sim \mathcal{M}. \tag{3}$$

Regarding the choice of objective function $L_\theta$ in continual learning, the simplest method is to directly apply standard loss functions from the IID setting, such as using cross-entropy loss for classification. In addition, many continual learning approaches also develop new loss functions to address catastrophic forgetting. One common technique is to use knowledge distillation, which involves adding a regularization loss term to explicitly preserve past knowledge (Li & Hoiem, 2017; Hou et al., 2019; Boschini et al., 2022).

### 3.2 HOW TO FAIRLY COMPARE ONLINE AND OFFLINE CONTINUAL LEARNING

**Computational cost.** Given epoch number $E$ and a task $C_i$ with a batch size of $B$, the number of gradient updates during offline CL on a task is $O(\frac{|C_i|}{B} \times E)$. In contrast, online rehearsal employs a single epoch setting. To deal with the challenge of underfitting in the single epoch setting, (Zhang et al., 2022) proposes to perform multiple gradient iterations for each incoming batch using Eq 3. Notably, with an iteration number $I > 1$, the same incoming batch $\mathcal{X}_t$ is reused in consecutive gradient updates. Since the memory batch is sampled from memory at each gradient update and only the incoming batch is reused, we called this partially biased SGD. Given an iteration number $I$ for each incoming batch and a data stream with a batch size of $B$, the number of gradient updates during online rehearsal on a task is $O(\frac{|D_t|}{B} \times I)$.

**Memory cost.** Apart from storing exemplars, offline continual learning also requires storing the whole task data for multi-pass training. Thus, the memory cost is $O(|M_{offline}| + |C_i|)$. In comparison, online CL does not store any task data and the memory cost is $O(|M_{online}|)$.

Space and compute complexity are highly important to continual learning. To enable a fair comparison between online and offline CL, we examine their performance under aligned memory and computational costs, formally defined as follows.

**Definition 6** (Aligned online and offline CL). Given a data stream and its task boundaries, consider an online continual learning algorithm $\mathcal{A}_{online}$ that performs gradient update Eq. 3 on a loss function $L_\theta$ for $I$ iterations per incoming batch and an offline continual learning algorithm $\mathcal{A}_{offline}$ that performs gradient update Eq. 2 on the same loss function $L_\theta$ for $E$ epochs per task. $\mathcal{A}_{online}$ and $\mathcal{A}_{offline}$ are aligned when $E = I$ and $|M_{online}| = |C_i| + |M_{offline}|$.

### 3.3 MAIN FINDING

**Align computation only**. We first investigate the effect of aligning computation only. Experiment setup can be found in Appendix C. Fig 1 shows when the iteration number is increased from 1 to 100, the performance of online ER is increased significantly. More interestingly, it becomes very close to the performance of offline ER, despite the fact online ER uses a much smaller memory budget: online-align-compute: 2064 samples (2k+64) and offline: 7000 samples (2k+5k)!

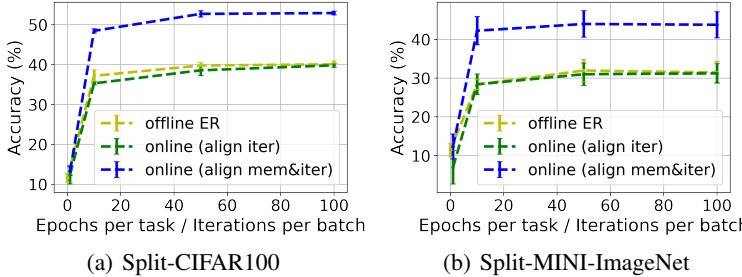

(a) Split-CIFAR100        (b) Split-MINI-ImageNet

Figure 1: Comparison of online and offline ER with aligned memory and computational cost. When aligning the iteration only, online ER achieves similar performance to its offline counterpart despite it requiring much smaller storage cost: offline 7k (2k+5k), online-align-iter 2.064k (2k+64). When aligning both the memory and iteration, online ER (7k) outperforms offline ER (7k) by a large margin.

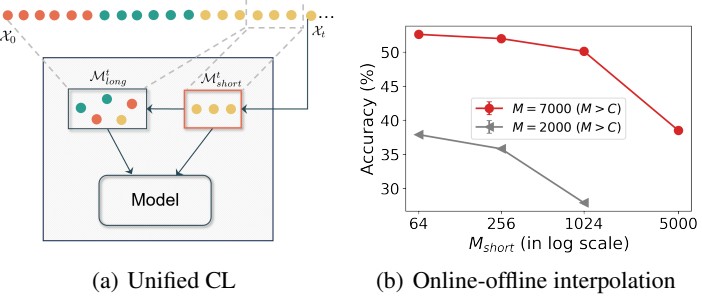

(a) Unified CL        (b) Online-offline interpolation

Figure 2: (a) Unified continual learning: online and offline CL are two special cases with $M_{short}$ equal to batch size in the online case and $M_{short}$ equal to task size in the offline case. (b) Performance results for a continuum of online and offline rehearsal, with online CL achieving the best performance.

**Align memory and computation.** With a total memory budget of 7k, Fig. 1 shows that online ER substantially outperforms offline ER under aligned iterations. This phenomenon is also observed in additional experiments on ImageNet (see Appendix D).

**Stability and plasticity**. Intuitively, for a given memory budget, online continual learning reserves more space to store past exemplars than offline continual learning, reducing forgetting. However, compared to unbiased SGD over multiple epochs in offline continual learning, the repeated usage of partially biased SGD in online continual learning may hamper learning efficacy on the current task, reducing plasticity. Thus, the comparison between aligned online and offline continual learning can be viewed as a tradeoff between stability and plasticity, with online continual learning trading plasticity for enhanced stability, and vice versa for offline continual learning.

Important open questions remain: 1) Why does online continual learning achieve a better stability-plasticity tradeoff than offline continual learning? 2) Does this online continual learning advantage persist across different problem settings with varying task and data stream sizes? 3) Is there a "sweet spot" between online and offline continual learning that optimizes the stability-plasticity tradeoff? We explore these questions in Sections 4 and 5.

## 4 A UNIFIED PERSPECTIVE OF ONLINE AND OFFLINE CONTINUAL LEARNING

### 4.1 UNIFIED CONTINUAL LEARNING

We revisit online and offline CL with aligned memory and computation, revealing that these two are the variants of the *exact same* algorithm with only difference in the storage allocated for the short-term memory. More specifically, we define a unified continual learning framework $UCL(M_{short}, M)$ (see definition 7). Given a total memory budget $M$, UCL maintains two memory buffers: a First-In-First-Our (FIFO) short-term memory to store $M_{short}$ recent samples, and a

long-term memory of size $M - M_{short}$ which stores a representative subset of samples excluding the $M_{short}$ recent ones. When the short-term memory is as large as the task size $\mathcal{C}_i$, $UCL(\mathcal{C}_i, M)$ is exactly offline CL. When the short-term memory is as large as the batch size $B$, $UCL(B, M)$ is online CL. Other special cases include IID training ($M_{short} = M = |\mathcal{D}_t|$) and rehearsal-free CL ($M_{short} = M = |\mathcal{C}_i|$).

**Definition 7** (Unified continual learning, $UCL(M_{short}, M)$). Given a data stream $\mathcal{D}_t = \cup_t \mathcal{X}_t$, a loss function $L_\theta$, and a total memory budget $M = |\mathcal{M}_{\text{long}}| + |\mathcal{M}_{\text{short}}|$. A short-term memory $\mathcal{M}_{\text{short}}$ uses a FIFO sliding window over the $M_{short}$ most recent batches, storing all samples: $\mathcal{M}_{\text{short}} = \cup_{t-M_{short}+1,...,t} \mathcal{X}_t$. A long-term memory $\mathcal{M}_{\text{long}}$ uses reservoir sampling to store a subset of the full data stream excluding the $N$ recent batches: $\mathcal{M}_{\text{long}} = \subset_{rs} \cup_{1,...,t-M_{short}} \mathcal{X}_t$. The gradient-based update rule is defined as:

$$\theta \leftarrow \theta - \eta \nabla L(\mathcal{B}_{long} \cup \mathcal{B}_{short}; \theta), \quad \text{where } \mathcal{B}_{long} \sim \mathcal{M}_{long}, \ \mathcal{B}_{short} \sim \mathcal{M}_{short}. \tag{4}$$

The training procedure of unified continual learning is outlined in Algorithm 1, with three key details: a) Instead of directly taking the incoming data stream batches, $\mathcal{M}_{\text{long}}$ processes ejected samples from $\mathcal{M}_{\text{short}}$ (line 18). b) Training begins only once $\mathcal{M}_{\text{short}}$ is full (lines 12-13), compensating with $\mathcal{M}_{\text{short}} \times I$ iterations initially (lines 15-16). c) $\mathcal{M}_{\text{short}}$ is emptied at the start of each new task, accommodating the offline CL approach. Ejected samples are then fed into $\mathcal{M}_{\text{long}}$ (lines 7-10).

### 4.2 STORAGE ALLOCATION POLICY

**Storage allocation policy**. Consider the unified continual learning framework with a total storage space $M$. An interesting question is how to allocate the space between short-term memory and long-term memory. We denote the storage allocation policy by a hyperparameter $\alpha = \frac{M_{short}}{M}$, which is the fraction of space allocated by the short-term memory.

**Semi-online CL**. Given the batch size $B$ and task size $C$ of the data stream, online CL always uses $\alpha = \frac{B}{M}$ and offline CL always uses $\alpha = \frac{C}{M}$. However, there is a wide range in between that has not been explored in the current practice of the field. Intuitively, $\alpha$ is related to the tradeoff between stability and plasticity. Saving more recent samples (higher $\alpha$) leads to better plasticity but worse stability and vice versa. For different CL problems and algorithms, this parameter of $\alpha$ can be tuned to find the one that works best. Fig 2 (b) shows the performance for a continuum of online and offline CL. Surprisingly, as $\alpha$ decreases, performance *monotonically* increases, achieving maximal accuracy with online CL. This experiment implies that online CL achieves the best performance along the online-offline continuum.

## 5 THEORETICAL ANALYSIS

### 5.1 GENERALIZATION BOUND

This section provides an analysis of the generalization error using the unified framework. To capture the influence of the non-stationary data distribution in continual learning, we leverage the concept of discrepancy distance from the transfer learning literature (Mansour et al., 2009). Discrepancy distance measures the distance between two distributions over a loss function $L$.

**Definition 8** (Discrepancy distance). Let $H$ be a set of functions mapping $X$ to $Y$ and let $L : Y \times Y \to R^+$ define a loss function over $Y$. The discrepancy distance between two distributions $Q_1$ and $Q_2$ over $X$ is defined by

$$\text{disc}_L(Q_1, Q_2) \doteq \max_{h,h' \in H} |\mathcal{L}_{Q_1}(h', h) - \mathcal{L}_{Q_2}(h', h)|.$$

where the expected loss of two functions over a distribution is denoted as $\mathcal{L}_{\mathbb{Q}}(f, g) \doteq \mathrm{E}_{x \sim \mathbb{Q}}[L(f(x), g(x))]$.

Let $\mathbb{D}$ and $\mathbb{M}$ denote the expected probability distributions of the data stream and the stored samples respectively. Let $\hat{\mathbb{M}}$ denote the empirical distribution of stored samples with a finite sample size of $M$. The true labeling function is defined as $h_y$. Given the optimal solutions $h_{\mathbb{M}}^* \doteq \operatorname{argmin}_{h \in H} \mathcal{L}_{\mathbb{M}}(h, h_y)$ and $h_{\mathbb{D}}^* \doteq \operatorname{argmin}_{h \in H} \mathcal{L}_{\mathbb{D}}(h, h_y)$, we present the generalization

---

**Algorithm 1:** Unified continual learning $UCL(M_{short}, M)$ (using ER as an example)

```
    // M is the total space budget of 𝓜_long ∪ 𝓜_short
    // M_short ∈ (batch size,task size)
 1  function TrainingEpochER(I, θ, 𝓜_short, 𝓜_long)
 2      for i = 1,...,I do
 3          𝓑_long ~ M_long, 𝓑_short ~ M_short
 4          θ ← θ − η∇L(𝓑_short ∪ 𝓑_long; θ)
 5      return θ

 6  function Unifed Continual Learning(𝒳_t, θ, 𝓜_short, 𝓜_long, n)
        // n is current sample numbers in 𝓜_short
 7      if 𝒳_t is a task boundary then
 8          𝓜_long ← ReservoirSampling(𝓜_long, 𝓜_short)
 9          𝓜_short ← ∅
10          n ← 0
11      𝓜_short, 𝒳_eject ← SlidingWindow(𝓜_short, 𝒳_t)
12      if 𝓜_short is not full then
13          n ← n + 1
14      else
15          if 𝒳_eject == ∅ then
16              θ ← TrainingEpochER(K × n, θ, 𝓜_short, 𝓜_long)
17          else
18              𝓜_long ← ReservoirSampling(𝓜_long, 𝒳_eject)
19              θ ← TrainingEpochER(K, θ, 𝓜_short, 𝓜_long)
20      return θ, 𝓜_short, 𝓜_long, n
```

---

bound during the long-short-term continual learning in Theorem 1. The proof is based on theorem 8 and proposition 2 in Mansour et al. (2009) (see Appendix A.1 for more details.)

**Theorem 1** (Generalization bound). Let $H$ be a hypothesis set bounded by some $A_0 > 0$ for the loss function $L : L(h, h') \leq A_0$, for all $h, h' \in H$. Assume that the loss function $L$ is symmetric and obeys the triangle inequality. Then, for any $h \in H$ and any $\delta > 0$, with probability at least $1 - \delta$, the following generalization bound holds:

$$\mathcal{L}_{\mathbb{D}}(h, h_y) \leq \mathcal{L}_{\hat{\mathbb{M}}}(h, h^*_{\mathbb{M}}) + \widehat{\Re}_{\mathcal{M}}(H) + 3A_0\sqrt{\frac{\log\frac{2}{\delta}}{2M}} + \text{disc}_L(\mathbb{D}, \mathbb{M}) + \mathcal{L}_{\mathbb{M}}(h^*_{\mathbb{M}}, h^*_{\mathbb{D}}) + \mathcal{L}_{\mathbb{D}}(h^*_{\mathbb{D}}, h_y),$$

(5)

where $\widehat{\Re}_{\mathcal{M}}(H)$ is the empirical Rademacher complexity of a hypothesis set $H$ over a sample set $\mathcal{M}$.

Considering the high expressive capacity of deep networks, we can assume the expected losses $\mathcal{L}_{\mathbb{M}}(h^*_{\mathbb{M}}, h^*_{\mathbb{D}})$ and $\mathcal{L}_{\mathbb{D}}(h^*_{\mathbb{D}}, h_y)$ are approximately zero. A key conclusion from Theorem 1 is that the generalization bound will be determined by the discrepancy distance $\text{disc}_L(\mathbb{D}, \mathbb{M})$ between the true data stream distribution $\mathbb{D}$ and the expected distribution $\mathbb{M}$ of the stored memory samples. Crucially, minimizing discrepancy distance $\text{disc}_L(\mathbb{D}, \mathbb{M})$ leads to a lower generalization bound.

## 5.2 OPTIMAL STORAGE ALLOCATION POLICY

To reduce the discrepancy distance $\text{disc}_L(\mathbb{D}, \mathbb{M})$, we investigate the influence of storage allocation policies. More specifically, we examine the effect of $\alpha$ on the discrepancy distance in Proposition 1.

**Proposition 1.** Assume $\mathbb{P}_+$ denotes the true probability distribution of the most recent task $\mathcal{C}_i$ and $\mathbb{P}_-$ denotes the true data distribution of all past tasks $\cup_{1,...,i-1}\mathcal{C}$. Given the number of samples seen in the data stream $N = \sum_t |\mathcal{X}_t|$ and the number of samples seen in the previous tasks $N^- =$

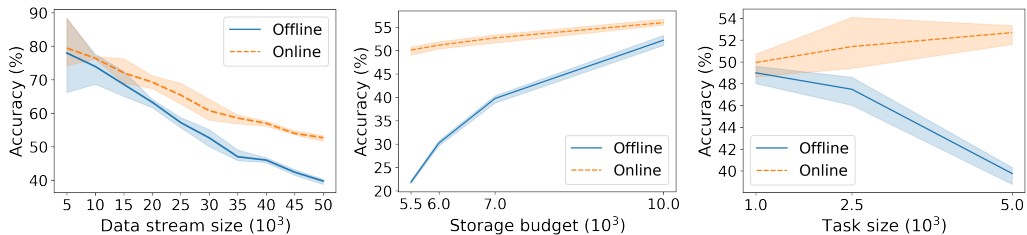

Figure 3: Performance difference between online and offline rehearsal in different problem settings: the online-offline performance gap becomes larger with longer data stream sequence $N$, smaller storage budget $M$, and larger task data size $C$.

$\sum_{k=1}^{k=i-1} |\mathcal{C}_k|$, we have:

$$\text{disc}_L(\mathbb{D}, \mathbb{M}; \alpha) = \left( \frac{N^-}{N} - \frac{(1-\alpha)N^-}{N - \alpha M} \right) \text{disc}_L(\mathbb{P}_-, \mathbb{P}_+). \tag{6}$$

Full proof is presented in Appendix A.2.

Proposition 1 characterizes how the discrepancy distance $\text{disc}_L(\mathbb{D}, \mathbb{M})$ qualitatively depends on task similarity $\mathbb{P}_-$ and $\mathbb{P}_+$, loss function $L$, and storage allocation factor $\alpha$. In the IID setting (i.e. $\mathbb{P}_- = \mathbb{P}_+$) or with a loss that leads to $\text{disc}_L(\mathbb{P}_-, \mathbb{P}_+) = 0$, we have $\text{disc}_L(\mathbb{D}, \mathbb{M}) = 0$ regardless of $\alpha$. However, when $\text{disc}_L(\mathbb{D}, \mathbb{M}) \neq 0$, the storage allocation factor $\alpha$ has a non-trivial effect. Combining Proposition 1 and Theorem 1 reveals the interplay between memory allocation, task similarity, and generalization capability in continual learning. In particular, when tasks are dissimilar and the loss cannot trivially minimize discrepancies, the allocation of storage between short-term and long-term memory impacts generalization performance. This implies the storage strategy is an important factor in continual learning when facing non-stationary data.

**Corollary 1.** When $\text{disc}_L(\mathbb{P}_-, \mathbb{P}_+) \neq 0$, the discrepancy distance $\text{disc}_L(\mathbb{D}, \mathbb{M}; \alpha)$ is minimized when $\alpha$ is minimal

$$\arg\min_{\alpha} \text{disc}_L(\mathbb{D}, \mathbb{M}; \alpha) = \alpha_{min} = \frac{B}{M}.$$

*Proof.* When $N > M$ and $\text{disc}_L(\mathbb{P}_-, \mathbb{P}_+) > 0$, we have $\nabla_\alpha \text{disc}_L(\mathbb{D}, \mathbb{M}; \alpha) = \frac{N^-(N-M)}{(N-\alpha M)^2} \text{disc}_L(\mathbb{P}^-, \mathbb{P}^+) > 0, \forall \alpha \in [\frac{B}{M}, \frac{C}{M}]$. $\square$

Corollary 1 shows that the optimal storage allocation strategy is to minimize the short-term memory, i.e. online CL. This leads to a lower generalization bound. Our analysis provides a theoretical explanation for the empirical finding that online CL achieves the best performance along the online-offline continuum (Fig. 2).

### 5.3 COMPARISON OF ONLINE AND OFFLINE CL IN DIFFERENT PROBLEM SETTINGS

This section examines the effectiveness of online and offline continual learning under different problem settings and studies what types of CL problems benefit most from online approaches.

**Corollary 2.** Given the data steam size $N$ with a batch size of $B$, a memory budget $M$, the data size of the most recent task $C$, the gap between the online and offline generalization bounds is:

$$R_L(N, M, C) \doteq \text{disc}_L(\mathbb{D}, \mathbb{M}_{offline}) - \text{disc}_L(\mathbb{D}, \mathbb{M}_{online}) = \frac{C - B}{N - B} \times \frac{N - M}{M} \text{disc}_L(\mathbb{P}^-, \mathbb{P}^+). \tag{7}$$

Corollary 2 reveals three key insights into the factors affecting the online versus offline rehearsal gap: 1) The advantage of online rehearsal increases as more data arrives ($\partial R_L / \partial N > 0$); 2) The gap diminishes with larger memory ($\partial R_L / \partial M < 0$), converging as $M \to N$. 3) Online rehearsal becomes more crucial as task size grows relative to stream frequency ($\partial R_L / \partial C > 0$).

We verify these dependencies empirically on Split-CIFAR100. Fig. 3 demonstrates that the online advantage expands when more data is seen, memory is limited, and tasks are larger, aligning with the theoretical results.

Table 1: Accuracy of aligned online and offline continual learning methods with 50 iterations/epochs. Offline CL always uses 2k exemplar. The exemplars of online is 2k + task size-batch size. $\alpha_{online}$ = batch size/memory budget and $\alpha_{offline}$ = task size/memory budget.

|  |  | ER | iCARL | DER++ | SCR |
|---|---|---|---|---|---|
| S-CIFAR100-10 | $\alpha_{\text{OFFLINE}}$ : 28.6% | $39.8 \pm 0.7$ | $47.7 \pm 0.4$ | $45.0 \pm 1.7$ | $47.2 \pm 0.6$ |
|  | $\alpha_{\text{ONLINE}}$ : 0.9% | $52.7 \pm 0.8$ | $49.7 \pm 0.3$ | $55.6 \pm 1.0$ | $56.3 \pm 0.1$ |
| S-MINI-IMAGENET-10 | $\alpha_{\text{OFFLINE}}$ : 28.6% | $31.9 \pm 2.8$ | $37.6 \pm 1.8$ | $23.1 \pm 4.1$ | $45.6 \pm 0.1$ |
|  | $\alpha_{\text{ONLINE}}$ : 0.9% | $43.9 \pm 3.4$ | $41.3 \pm 2.6$ | $42.0 \pm 4.8$ | $51.3 \pm 0.5$ |
| S-CORE-9 | $\alpha_{\text{OFFLINE}}$ : 85.7% | $40.1 \pm 2.4$ | $45.8 \pm 1.6$ | $22.9 \pm 4.4$ | $62.1 \pm 1.3$ |
|  | $\alpha_{\text{ONLINE}}$ : 0.0% | $50.7 \pm 1.7$ | $50.1 \pm 1.6$ | $46.7 \pm 2.6$ | $69.7 \pm 0.3$ |

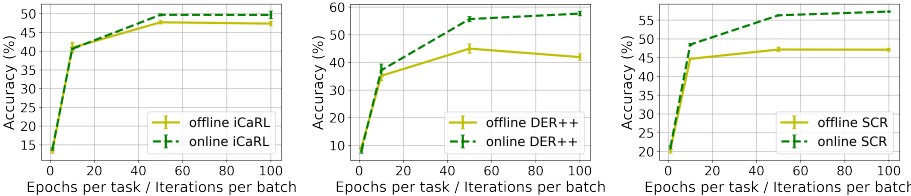

Figure 4: Online and offline comparison with CL algorithms.

# 6 ADDITIONAL EXPERIMENTS

**Knowledge distillation**. To reduce forgetting, many CL techniques leverage knowledge distillation to construct a regularization loss, appointing a past model as the teacher and current model as the student (Li & Hoiem, 2017; Rebuffi et al., 2017; Buzzega et al., 2020; Hinton et al., 2015). We investigate the interplay between storage allocation parameters with the knowledge distillation mechanism using iCaRL (Rebuffi et al., 2017) and DER++ (Buzzega et al., 2020)). Table 1 shows using a smaller storage allocation parameter $\alpha$ leads to performance improvement for both iCaRL and DER++ for three datasets, although the performance boost seems to be smaller in iCaRL.

**Contrastive learning**. Some recent works (Cha et al., 2021; Mai et al., 2021; Khosla et al., 2020) apply contrastive learning to mitigate forgetting and achieve state-of-the-art performance in both online and offline CL fields. We replace cross-entropy with contrastive loss and demonstrate that the storage allocation parameter also plays an important role when using a small $\alpha$ leads to better performance for SCR algorithm.

**Zero short-term memory.** In addition to online and offline CL, we considered the extreme case of having zero short-term memory, as explored in the Gdumb baseline (Prabhu et al., 2020). Our generalization analysis shows that zero short memory results in a lower bound than non-zero cases, providing some insight into Gdumb's strong performance. However, zero short-term memory leads to a significantly different training procedure compared to non-zero $M_{short}$. Specifically, with non-zero $M_{short}$, the model trains on each incoming batch before discarding it, regardless of $M_{short}$'s size. With zero $M_{short}$, incoming data goes straight into long-term memory via reservoir sampling, without model training. We conducted additional experiments with zero $M_{short}$ and found it underperforms small non-zero $M_{short}$ (e.g. online ER 52.7% vs zero-ER 50.3%, online SCR 56.3% vs. zero-SCR 52.5%). Therefore, when considering zero $M_{short}$, it is important to account for the alterations to the training procedure compared to standard continual learning systems.

# 7 CONCLUSION

Many continual learning techniques are applied offline in a task-based manner. Conventional wisdom holds that these methods may fail when applied in an online manner due to catastrophic forgetting and underfitting from single-pass data. This paper challenges that assumption by empirically showing comparable or better performance for online task-free learning given equal memory and computational resources. We corroborate these experiment findings by a systematical theoretical construction showing online and offline CL can be unified via a long-short-term memory framework and online CL yields better generalization bound. By fundamentally rethinking the comparison between online and offline continual learning, we hope this work stimulates further research into efficient lifelong learning algorithms capable of effectively mastering new skills over time.

**Limitations**. This work examines rehearsal and regularization techniques for continual learning. Other approaches, such as correcting task recency bias (Wu et al., 2019; Hou et al., 2019) or expanding network capacity (Zhou et al., 2022; Yan et al., 2021), are not covered by our analysis. Additionally, we assume random reservoir sampling for exemplar selection. Alternative strategies to construct representative data summaries (Borsos et al., 2020; Bang et al., 2021) may interact differently with the online-offline continuum, presenting another area for empirical analysis through the proposed framework.

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

# A PROOFS

## A.1 PROOF OF THEOREM 1

*Proof.* Based on triangle inequality and the definition of discrepancy distance (theorem 8 in Mansour et al. (2009))

$$\mathcal{L}_{\mathbb{D}}(h, y) \leq \mathcal{L}_{\mathbb{D}}(h_{\mathbb{D}}^*, y) + \mathcal{L}_{\mathbb{M}}(h, h_{\mathbb{M}}^*) + \mathrm{disc}_L(\mathbb{D}, \mathbb{M}) + \mathcal{L}_{\mathbb{M}}(h_{\mathbb{M}}^*, h_{\mathbb{D}}^*). \tag{8}$$

Following the property of Rademacher Bound (proposition 2 in Mansour et al. (2009)), we have

$$\mathcal{L}_{\mathbb{M}}(h, h_{\mathbb{M}}^*) \leq \mathcal{L}_{\hat{\mathbb{M}}}(h, h_{\mathbb{M}}^*) + \widehat{\mathfrak{R}}_{\mathcal{M}}(H) + 3C \sqrt{\frac{\log \frac{2}{\delta}}{2M}} \tag{9}$$

Inserting Eq 9 into Eq 8 gives theorem 1. □

## A.2 PROOF OF PROPOSITION 1

*Proof.* Let $\gamma \doteq \frac{N^-}{N}$. Based on the definition of discrepancy distance, we have:

$$\begin{aligned}
\mathrm{disc}_L(\mathbb{D}, \mathbb{M}; \alpha) = \max_{h, h' \in H} |\gamma \mathcal{L}_{\mathbb{P}^-}(h', h) + (1 - \gamma)\mathcal{L}_{\mathbb{P}^+}(h', h) \\
- \left((1 - \alpha)\mathcal{L}_{\mathbb{M}_{long}}(h', h) + \alpha \mathcal{L}_{\mathbb{M}_{short}}(h', h)\right)|
\end{aligned} \tag{10}$$

Since the long-term memory is managed by the reservoir sampling method and the short-term memory is managed by the sliding window method, and letting $\beta \doteq \frac{N^-}{N - \alpha M}$, we have $\mathcal{L}_{\mathbb{M}_{long}}(h', h) = \beta \mathcal{L}_{\mathbb{P}^-}(h', h) + (1 - \beta)\mathcal{L}_{\mathbb{P}^+}(h', h)$ and $\mathcal{L}_{\mathbb{M}_{short}}(h', h) = \mathcal{L}_{\mathbb{P}^+}(h', h)$. Inserting these two results into Eq 10 gives:

$$\begin{aligned}
\mathrm{disc}_L(\mathbb{D}, \mathbb{M}; \alpha) &= \max_{h, h' \in H} |(\gamma - (1 - \alpha)\beta)) \left(\mathcal{L}_{\mathbb{P}^+}(h', h) - \mathcal{L}_{\mathbb{P}^-}(h', h)\right)| \\
&= (\gamma - (1 - \alpha)\beta)) \max_{h, h' \in H} |\left(\mathcal{L}_{\mathbb{P}^+}(h', h) - \mathcal{L}_{\mathbb{P}^-}(h', h)\right)|
\end{aligned} \tag{11}$$

This last equality is based on the fact that $\gamma - (1 - \alpha)\beta = \frac{\alpha N^-(N - M)}{N(N - \alpha M)} > 0$ when $N > M$. □

# B DATASET DETAILS

Table 2 lists the image size, the number of classes, the number of tasks, and data size per task of the four CL benchmarks.

Table 2: Dataset information for the four CL benchmarks.

|  | IMAGE SIZE | #TASK | # CLASS | TRAIN PER TASK | TEST PER TASK |
|---|---|---|---|---|---|
| SPLIT-CIFAR100 | 3x32x32 | 10 | 100 | 5,000 | 500 |
| SPLIT-MINI-IMAGENET | 3x84x84 | 10 | 100 | 5,000 | 1,000 |
| SPLIT-CORE50-NC | 3x128x128 | 9 | 50 | 12,000 | 4,500 |
| SPLIT-IMAGENET-1K | 3x224x224 | 10 | 1000 | $\sim$120,000 | 5000 |

## C  EXPERIMENT DETAILS

**Experiment setup.**  We empirically compare aligned online and offline CL on two popular CL benchmarks, Split-CIFAR100-10 and Split-Mini-ImageNet. Both consists of 10 tasks, with each task containing 5000 images. For offline rehearsal, we allocate 2000 exemplars to the memory buffer. Following Definition 6, the aligned online rehearsal setting reserves 7000 exemplars. We evaluate four computation budgets defined by offline epoch count or online iteration count: 1, 10, 50, and 100. Cross-entropy loss is used in this experiment. Following Masana et al. (2022), all experiments utilized ResNet-18 with a single head, standard data augmentation (random cropping and flipping), and a batch size of 64. We report the end accuracy after completing all tasks.

**Hyperparameters**. The hyperparameter settings are summarized in Table 3. The regularization strength in DER++ and temperature values in SCR follow original papers. Each training batch contains 64 new samples and 64 memory samples. All models use vanilla SGD for optimization. For iCaRL and SCR, a Nearest-Class-Mean (NCM) classifier is applied as in the original works. The default iteration and epoch number is 50. We run all experiments across three random seeds.

Regarding the metrics, the performance of CL is measured by the end accuracy after training on all tasks, defined as $A_T = \frac{1}{T}\sum_{j=1}^{j=T} a_{T,j}$, where $a_{i,j}$ denotes the model's accuracy on the held-out test set of task $j$ after training on task $i$. Other metrics are "forgetting" (Chaudhry et al., 2018), which is defined as $F_T = -\frac{1}{T-1}\sum_{i=1}^{T-1}\left(a_{T,i} - \max_{l\in 1...T-1} a_{l,i}\right)$ and the related metric "backward transfer" (Lopez-Paz & Ranzato, 2017): $B_T = \frac{1}{T-1}\sum_{i=1}^{T-1} a_{T,i} - a_{i,i}$. And the stability and plasticity is defined as follows Zhang et al. (2022):

$$A_T = \underbrace{\frac{1}{T}\Sigma_{i=1}^T a_{i,i}}_{\text{Plasticity}} + \underbrace{\frac{T-1}{T}B_T}_{\text{Stability}} \geq \frac{1}{T}\Sigma_{i=1}^T a_{i,i} - \frac{T-1}{T}F_T.$$

Table 3: Hyperparameter setting.

|  | HYPERPARAMETER |
|---|---|
| ER | LR=0.1 |
| iCaRL | LR=0.1,NCM CLASSIFIER |
| SCR | TEMP =0.07, LR=0.1, NCM CLASSIFIER |
| DER ++ | $\alpha = 0.1$, $\beta = 0.5$,$lr = 0.03$ (CIFAR100) |
|  | $\alpha = 0.3$ $\beta = 0.8$,$lr = 0.1$ (MINI-IMAGENET) |
|  | $\alpha = 0.1$, $\beta = 1.0$,$lr = 0.1$ (CORE50) |

## D  ADDITIONAL EXPERIMENT RESULTS

Additional experiment results on Split-Mini-ImageNet100-10 is shown in Fig D and the experiment result on Split-ImageNet1000-10 is shown in Table 4. Both experiments confirm the effectiveness of online rehearsal over offline rehearsal.

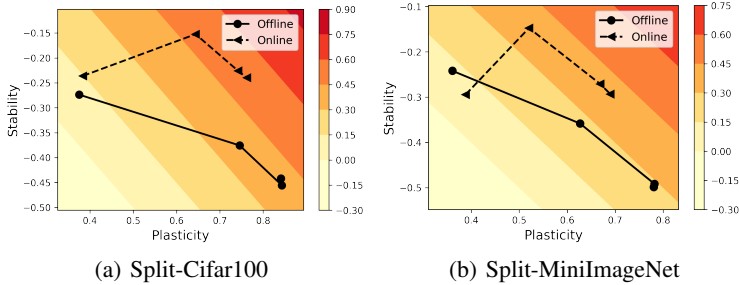

(a) Split-Cifar100 (b) Split-MiniImageNet

Figure 5: Stabiltiy and plasticity during online and offline ER with increasing iterations/Epochs.

Table 4: Accuracy of aligned online and offline continual learning methods in ImageNet1000 with an incremental of 100 classes. Offline ER uses 20,000 exemplars and online ER uses 120,000 exemplars. Due to the computation time constraint, we report only the results of the first 500 classes.

|  | 100 CLASSES | 200 CLASSES | 300 CLASSES | 400 CLASSES | 500 CLASSES |
|---|---|---|---|---|---|
| ER OFFLINE | 81.6 | 50.4 | 48.0 | 43.1 | 39.4 |
| ER ONLINE | 82.5 | 70.2 | 62.0 | 60.0 | 58.5 |

