# OpenReview forum: "Bridging the gap between offline and online continual learning"
_ICLR.cc/2024/Conference — Submitted to ICLR 2024_

### Official Review · Reviewer_LdF8 · 2023-10-26

**Soundness:** 3 good
**Presentation:** 3 good
**Contribution:** 2 fair
**Rating:** 3
**Confidence:** 5

**Summary:**

The authors compare the *online learning* and *offline learning* setting by proposing to allow for the *online learning* setting as much memory as in the *offline learning* setting. They provide detailed theoretical and experimental analysis of multiple memory allocation settings, thus effectively interpolating between the online setting and the offline setting. Experiments are done on Split-Cifar100, Mini-Imagenet100 and Core50, and using several competitve methods like ER, SCR and ICARL. They find that under a fixed memory budget, it is more efficient to store most of the memory in the long term memory rather than in the short term memory (thus, the *online setting* is more efficient).

**Strengths:**

- It's an interesting idea to compare the two settings under these circumstances. It is true that the common definition does not impose constraints on the computational cost, so it makes sense to compare it's performance to the one of offline learning under the same computational cost
- The discovery that less short term memory is better is interesting and could help the field to move forward

**Weaknesses:**

- 1) While some theoretical analysis is provided, I don't think that this is sufficient to prove the claims of the paper. In particular, the analysis of Eq (6) seem to show that "a minimum amount of short term memory" should be used. In that case, why are you using short term memory at all if the analysis concludes that you should not use it ? (It's also a question, please inform me if I missed something here). In the case that I am correct, I think that these analysis are not necessary to the paper.
- 2) I think an important comparison is missing to the paper which is for me the main point. You claim that the optimal setting is using a small short-term memory but I have not seen experiments not using any short term memory (which would collapse to the GDumb baseline [1]). Indeed, if this baseline performs well under your experimental setting then the claim of the paper changes completely and the novelty is not here anymore. So I think it's very important to include this baseline that uses all the memory available (i.e 7k for cifar100) as long term memory.
- 3) It would be nice to have results for Split-Cifar100 and Mini-Imagenet with more splits (20), since then the total allowed memory would also be smaller.

[1] "GDumb: A Simple Approach that Questions Our Progress in Continual Learning", A Prabhu, PHS Torr, PK Dokania

**Questions:**

- Do you plan to release the code ?
- c.f weaknesses (2), did I understand something wrong in the analysis Eq (6) ? If the analysis concludes that you should use minimum short term memory why do you use any at all ? If it's just that it's working better experimentally but that the analysis concludes something different then I think it's better to remove this analysis altogether.

---

> ### Author Response · Authors · 2023-11-21
> **Author reply to Reviewer LdF8**
>
> We appreciate the reviewer constructive feedback and suggestions. We address the concerns and questions as below:
>
> **W1. Why we argue to use the minimum short-term memory instead of zero short memory**
> >the analysis of Eq (6) seem to show that "a minimum amount of short term memory" should be used. In that case, why are you using short term memory at all if the analysis concludes that you should not use it?
>
> Great question. The reason is that $M_{short}=0$ leads to a significantly different procedure with non-zero short memory. More specifically,  for any non-zero short-term memory, no matter how small it is, there is a consistent training procedure that spans online and offline CL: each incoming batch will be trained by the model for $E$ iterations before it is discarded. Due to the continual learning nature, this procedure holds regardless of the exact value of $M_{short}$. In contrast, if there is no short-term memory, the incoming data is directly saved to long-term memory based on some selection mechanism _without being trained_. This means some incoming data are never seen by the model. We believe this represents a fundamentally distinct training approach from the current online and offline CL practices.
>
> Our theoretical analysis only concerns the generalization bound and does not cover training procedures. Since the performance is affected by both factors, we apply our theoretical results only to provide a possible explanation for algorithm performance with similar training procedures.  Hence, we interpret our theoretical results only in the context of long-short term frameworks, where short-term memory lies in $[DataStreamBatchSize, TaskSize]$.
>
>
> **W2.1 Using zero short memory as a baseline.**
>
> Thanks for this advice. We added a paragraph in the paper to discuss the zero short memory setting and its relation to Gdumb.
> With zero short term memory, the generalization bound analysis does not change much. Comparing $M_{short}=0$ and $M_{short}=DataStreamBatchSize$, with a small batch size, the generalization bounds are very similar, with the former leading to a slightly lower bound. However, the difference in training procedures must also be considered along with the generalization bounds. As shown in the table below, the performance of zero short memory is slightly worse than minimum short memory (ER 2.4% $\downarrow$, SCR 3.8% $\downarrow$), but still much better than $M_{short}=$ task size (offline CL). We believe this is the combined result of the generalization bounds and the training procedures.
>
> |                      | ER     |  SCR
> |----------------------|----------------| ---------------|
> | M_short: task size | 39.8 $\pm$  0.7 |  47.2 $\pm$  0.6 |
> | M_short: batch size| 52.7 $\pm$ 0.8 | 56.3 $\pm$  0.1|
> |  M_short: zero    | 50.3 $\pm$1.4  | 52.5 $\pm$ 0.4 |
>
>
>
> **W2.2: zero short memory is equal to Gdumb. Novelty is not here anymore**
>
> We would like to clarify that Gdumb and the zero-short memory setting (referred as zeroShort) investigated here have some fundamental differences and send different messages:
>
> 1) Every time the model is updated, the model is re-trained from scratch in Gdumb. On the contrary, zeroShort is a continual learner which updates model continually with E gradient steps when one incoming batch arrives.
> 2) Gdumb maintains a class-balanced memory while zeroShort uses reservoir sampling.
> 3) When compared with CL algorithms, Gdumb employs a separate learning design (cutmix augmentation, training epochs, retrain from scratch). Gdumb outperforms online CL in Table 3,4 in Gdumb paper [1]. Our experiment removes the compounding effect of learning algorithms and sends a different message: online-CL is a bit better than zeroShort-CL.
> 4) Gdumb reported worse results than offline CL algorithms (Table 5, B2 in [1]). Only the exemplar memory is aligned in the Gdumb experiments. We align the total memory budget and report zeroShort-CL is much better than offline CL. Thus, we believe Gdumb is undervalued in its comparison with offline CL.
>
>
> **W3. results with more splits and different total memory budget**
>
> The results of Split-Cifar100 with 50, 20 and 10 are shown in Fig 3 c. We use task size as the labels for x axis, i.e. 1.0 k, 2.5k, 5k. These experiments uses 7k total memory.
>
> Besides using 7k memory budget, Fig 3b experiments with other total memory budgets (5.5k, 6k, 7k, 10k )
>
> **Q1**. yes, we will release the code
>
> **Q2**. Hope the response above has addressed this question. Please inform us if you have any further inquiries.

---

> > ### Comment · Reviewer_LdF8 · 2023-12-04
> >
> > Thank you for your replies and for the additional experiments. After looking at the results of the comparison with GDumb, I am not so sure about the interest of the ideas developed in the paper. I initially thought that the results presented in the paper would be way better than the ones proposed by GDumb, however, I see that there is only a 2 to 4% difference, which in my opinion is not significative enough to conclude on the power of "learning online" with the same computational budget than offline learning. Also, I still strongly believe that the presence of an analysis that concludes that GDumb is the optimal setting and is followed by different experiments is problematic.
> >
> > For those reasons, I will decide to lower my rating

---

### Official Review · Reviewer_tpnv · 2023-10-27

**Soundness:** 3 good
**Presentation:** 3 good
**Contribution:** 2 fair
**Rating:** 5
**Confidence:** 4

**Summary:**

This paper empirically demonstrates that online CL can match or exceed the performance of its offline counterpart given equivalent memory and computational resources. Then the paper give a theoritical explanation from the optimal storage allocation policy.

**Strengths:**

1. the observation of online CL can match or exceed the performance of its offline counterpart given equivalent memory and computational resources is interesting.
2. This paper further verifies the observation on other methods.
3. The theoritical explanation from the optimal storage allocation policy is valid.

**Weaknesses:**

(1) It seems that Figure 1 and Figure 1 are the same figure. You need to fix this problem.
(2) The experiment for supporting the emprical observation is not very valid. You should consider more complex datasets (e.g., ImageNet) or class incremental learning in NLP tasks (e.g., intent classfication). Only conducting experiments on the CIFAR dataset is not enough.

(3) I don't know what benifit can we learn from this observation? Can we use it to  reduce the forgetting problem?

**Questions:**

Can we use it to improve the performance of current online continual learning methods? Can it principally reduce the forgetting of vision or NLP models in continual learning? I do not see the benifit of this observation.

---

> ### Author Response · Authors · 2023-11-22
> **Author response to Reviewer tpnv**
>
> Thank you for taking the time to review our paper and provide feedback. We address your concerns and questions point by point as follows:
>
> **W1. Figures**
>
> >(1) It seems that Figure 1 and Figure 1 are the same figure. You need to fix this problem.
>
> Figure 1 and Figure 5 are not the same figure. Figure 1 is the results on Split-CIFAR100 and Figure 5 are the results on Split-Mini-ImageNet
>
> **W2. Experiments beyond CIFAR**
> >The experiment for supporting the emprical observation is not very valid. You should consider more complex datasets (e.g., ImageNet) or class incremental learning in NLP tasks (e.g., intent classfication). Only conducting experiments on the CIFAR dataset is not enough.
>
> We would like to clarify that apart from Split-CIFAR100, we also conducted experiments in other widely used continual learning benchmarks, split-mini-imagenet and CORE50 (see Table 1).  The appendix also includes the results of ImageNet in Table 4.  Our finding is consistent in all these experiments with different image types and dataset size.
>
> **W3.1 The implication of our finding**
> >I don't know what benefit can we learn from this observation?
>
>  This work poses several questions into the current practice of the continual learning field:
> | Current pratice                                                                                                                                                                              | Our contribution                                                                                                                                                                 | Implications                                                                                                                                                              |
> |------------|--------------|-------------|
> | Focus on the storage cost of exemplar samples                                                                                                                                               | Exemplar and new samples both exist within the memory-constrained operational space; Take into account both in our experiment | Online CL is probably undervalued in the community and it may provide better solutions than offline CL  in a practical application with limited memory.|
> | Use single iteration in online CL (I=1)  and large epochs in offline CL (E=70-200); Complex offline CL algorithms are believed to be not suitable for online CL due to the single epoch constraint |  Experiment with the iteration number as large as the epoch number; Demonstrate complex algorithm design (e.g DER++) can be trained well in single epoch setting (online) by simply increasing iteration number |  Open the door of more algorithms candidates (the second line in the table below) for solving online CL problems                                                                                             |
>
>
>
> |            | Algorithms                                                                            |
> |------------|---------------------------------------------------------------------------------------|
> | online CL  | ER, ER-ACE, MIR, ASER, GSS, SCR, RAR, COPE, OCM,CBA                                             |
> | offline CL |  iCaRL, BIC, IL2M, LUCIR, EEIL, RWalk, DER, MEMO, FOSTER, PODNET, FOSTER, WA, RMM, COIL, CO2L, DER++|
>
> We revised the introduction and conclusion of the paper to clarify the implications of our work.
>
>
> **W3.2 Can the finding in this paper be used to reduce forgetting?**
>
> Yes. The stability-plasticity analysis in Figure 1(b) highlights the effect of the storage allocation policy on forgetting: using a small storage allocation parameter leads to better stability (less forgetting). One of our primary messages is that the storage allocation policy is a key factor concerning CL performance but is overlooked in the current practice of the field. Our experiment results on four CL algorithms suggest that tuning the storage allocation parameter leads to a bigger performance boost than specific algorithm design (Table 1).

---

### Official Review · Reviewer_QLLJ · 2023-10-30

**Soundness:** 1 poor
**Presentation:** 3 good
**Contribution:** 3 good
**Rating:** 3
**Confidence:** 4

**Summary:**

This work tries to bridge the gap between offline and online continual learning. It corrects the memory sizes and computational budgets to bridge this gap:

1) Issue in Memory:
- Offline CL – $M_{offline}$ has access to all samples of this task along with memory size of M, O(|M| + |$C_{task}$|)
- Online CL – $M_{online}$ stores has access to only the incoming batch along with memory size of M, O(|M| + |$C_{batch}$|).
- where $C_{batch}$ << $C_{task}$
- (Note: It’s not just |M| as claimed)

**Correction**: |$M_{online}$| = |$M_{offline}$ + $C_{task}$ - $C_{batch}$|

2) Issue in Computation:
- Offline CL approaches uses far more computation compared to Online CL methods!

**Correction**: Equalize both offline and online CL methods to use the same computational budget C.

*Experiments*: In Section 3.3 and 6, the paper shows that once both corrections are implemented, online continual learning methods outperform offline continual learning methods.

3) Then, the work focuses on interpolating between |$M_{offline}$| and |$M_{online}$| with short-term and long-term memories.

- Short-term memory ($M_{short}$) : FIFO ordering (stores last |$M_{short}$| samples), emptied at each task boundary.
- Long-term memory ($M_{long}$): Stores the latest sample ejected from the short-term memory by reservoir sampling mechanism.

They start with a theoretical analysis, showing effects and results obtained from minimizing discrepancy. Please let me know if there are inaccuracies in my summary.

**Strengths:**

[Computation] The correction to normalize computational costs is correct, and fleshing out findings based on that assumption further can be very valuable!

[Easy to Read] The paper was understandable, even though it took some time to navigate through the notations. Would have preferred an informal claim alongside the formal notations/theorems for building intuitions and ease-of-reading.

[Good contribution] This work investigates a valuable problem in my view!

This is a partial review, once my main concerns are addressed I can give detailed comments in the feedback period for further sections.

**Weaknesses:**

I am only listing critical issues in the work currently. If resolved, I can list the other issues in the work.

**W1.** *$C_{task}$ and $M_{online}$ have different distributional constraints* [**Critical**]
- The fundamental discrepancy arises from the proposed memory correction. Offline continual learning employing a 2000-sized memory buffer ($M_{offline}$) is notably disadvantaged relative to online continual learning with a 7000-sized memory ($M_{online}$).
- This stems from the fact that while the data distribution within a task ($C_{task}$) remains static, the memory ($M_{online}$) has the potential to represent samples from all past data distributions.
- Concretely, I think the primary factor favoring online CL over offline CL in Section 3.3 comes from this imbalance in representation. 5000 samples from the latest distribution $C_t$ (in $M_{offline}$) are markedly less useful than 5000 samples across $D_1, ..., D_t$ (in $M_{online}$), hence it performs poorer.

I would be very surprised if online continual learning ever outperforms the offline continual learning setting, elaborated below.

*What if , with the only difference being short-term memory?*

What is the fair correction to memory then? (I agree with the paper equalizing computation)
- Equalize the long-term memory alongside computation. Interpolate between short-term memory from size $C_{batch}$ to $C_{task}$

- Concretely, in Section 3.3, the offline model can use a far larger short term memory (5000) compared to an online model (64). Both have the same fixed sized long term memory but have equal computation.

I claim (informally): Any training algorithm (with compute budget C) where 64 samples are sequentially introduced can be mimicked when you have access to those 64 and subsequent samples (totaling 5000 samples). The reverse is not feasible due to the inaccessibility of future samples.

Conclusion: Offline continual learning should be strictly better than online continual learning in principle! (Strictly better because future samples help improve performance)

However, this diametrically opposite to the aim of this work. I hope the above informal statements conveys the core intuition behind why offline continual learning is going to be better than online alternatives.

**W2** *Bound not tight, and likely vacuous* [**Critical**]
- Theorem 1, in its current form, merely establishes an inequality without showing that the bound is tight. However, the subsequent arguments rely heavily on the bound being tight.
- For proof of theorem 1 to be complete, one must additionally show that the same hypothesis h* achieves equality in both Eq 9 and Eq 8 in Appendix A.1. It is unclear to me that the best hypothesis on distribution of memory samples will necessarily be the best for overall data distribution. However, note that I haven't looked at Mansour et al. (2009).

 *Would the tightness, if shown, convince me?*

- It would make the proof of the theorem complete. However, for deep learning systems, generalizations bounds likely more sophisticated than Theorem 1 are shown to be vacuous [1]. I would not hinge my entire motivation on minimizing discrepancy due to this.
- Because of this reason, I think the theoretical motivations for the empirical results are quite poor and Section 5 adds little value to the work.
- However, I am perfectly happy to consider predictions from Corollary 2.2 on their own terms and empirical results in Figure 3 from it.

[1] Uniform convergence may be unable to explain generalization in deep learning.

**Questions:**

Currently, the paper comprises of incorrect memory assumptions used for empirical results (W1: Section 3.3 and Section 6) and vacuous  theoretical motivations (W2: Section 5).

I am open to drastically changing my current score if **W1** is thoroughly addressed by the authors. However, as it stands I think there are fundamental flaws in the work due to confusing the nature of $C_{task}$ and $M_{online}$ and I am fairly confident in this claim.

---

> ### Author Response · Authors · 2023-11-17
> **Author Response to W1 raised by Reviewer QLLJ (Part I)**
>
> We greatly appreciate the reviewer providing their main concern W1 in a partial review to engage in further discussion. For expediency, we focus on replying to this concern here and will upload our replies to other concerns and the revised paper later.
>
> ## The effect of aligning computation only
>
> In our submission, we only presented the results of aligning both memory consumption and the number of iterations. Prompted by W1, we have now included an ablation study with results obtained by aligning the number of iterations or memory consumption only, summarized in the table below. Compared with no alignment of the number of iterations, regardless of whether memory consumption is aligned or not, the performance of online CL is significantly increased by moving from 1 to 50 iterations. More interestingly, when aligning the number of iterations only, the performance of online CL  becomes very close to the performance of offline CL, despite the fact it uses a much smaller memory budget: online-align-compute: 2064 samples (2k+64) and offline: 7000 samples (2k+5k)!
>
>
>
> |                           | #Iterations   |Memory    |CIFAR100    | MINI-IMAGENET |
> |---------------------------|-------------|---------------|---------------|---------------|
> | offline   | 50 | 7K|40.1 $\pm$ 1.6 |    31.9 $\pm$ 2.8           |        |
> | online-no-align | 1 | 2.064K  | 11.6 $\pm$ 1.1 | 11.4 $\pm$ 1.8              |        |
> | online-align-compute |50|2.064K | 38.6 $\pm$ 1.4 |    31.0 $\pm$ 2.9          |        |
> | online-align-memory   |   1 | 7k | 13.2   $\pm$ 1.2        |   12.3 $\pm$ 3.2            |        |
> | online-align-both  |50|7K        | 52.7 $\pm$  0.8 | 43.9 $\pm$3.4
>
> Thanks for pointing out our error in calculating the memory cost of online CL. We updated it to exemplar memory + batch size.

---

> > ### Author Response · Authors · 2023-11-17
> > **Author Response to W1 raised by Reviewer QLLJ (Part II)**
> >
> > ## Why we choose to align the memory budget
> >
> >
> > > W1.  $C_{task}$ and $M_{online}$ have different distributional constraints [Critical]...Concretely, I think the primary factor favoring online CL over offline CL in Section 3.3 comes from this imbalance in (data distribution) representation.
> > 5000 samples from the latest distribution $C_t$ are markedly less useful than 5000 samples across all previous datasets $D1,...,Dt$
> >  , hence it performs poorer.
> >
> > We completely agree with the reviewer that with an aligned memory budget, online and offline CL have different distributional constraints: the primary factor favoring online CL is that it is able to store fewer samples from the latest distribution and more samples from past distributions. It seems that this simple observation has been overlooked in prior work on CL. However, we strongly disagree that this means experiments with aligned memory are not worth performing. On the contrary, aligning memory budget provides a vital lens for studying efficient storage distribution constraints and representation mechanisms for CL. More specifically, 1) aligned memory allows systematic comparison of  the usefulness of recent samples and past samples, and  2) it reveals the effect of the storage allocation mechanism for balancing samples from past and present on the performance of continual learning.
> >
> > **1) Past vs. present: on the ``usefulness`` of past and recent samples**
> >
> > > 5000 samples from the latest distribution $C_t$ are markedly less useful than 5000 samples across all previous datasets $D_1$,...$D_t$"
> >
> > Does this statement always hold, regardless of data stream size, memory size, or CL algorithm (ER, SCR, iCaRL,DER etc)? And if so, why? These are the exact questions we attempt to answer in this paper, and based on our understanding, this is not straightforward. On the one hand, training on samples from the latest distribution enhances plasticity, and training on samples from the past enhances stability. On the other hand, samples from the past have already been seen by the model and may provide a poorer signal for learning than recent ones. Additionally, CL algorithms may leverage past samples and present samples through different loss designs and training mechanisms to consolidate existing knowledge and mitigate forgetting. This interplay renders the above questions non-trivial; the answers are important to memory-based CL and cannot be found in the literature to our knowledge.
> >
> >  To investigate these questions, we study, given a total memory budget $M$, how different ratios of short-term and long-term memory affect CL performance.
> >  By studying the effect of the storage allocation parameter $\alpha=\frac{M_{short}}{M}$ we show that the usefulness of recent samples and past samples may vary with the problem structure (i.e., data stream size, memory budget; see fig 3) and loss functions (see fig 4).
> >
> >
> >
> > **2) On the importance of storage allocation parameter $\alpha=\frac{M_{short}}{M}$**
> >
> > One of our primary messages is that the storage allocation policy is a key factor concerning CL performance and should be considered in a more principled manner. And our experiment results on four CL algorithms suggest that tuning the storage allocation parameter $\alpha$ leads to a bigger performance boost than specific algorithm design (Table 1).
> >
> > Online and offline CL can be viewed as variants of the same learning algorithm that only differ in the setting of the storage allocation parameter (Online CL always uses the minimum  $\alpha=\frac{DataStreamBatchSize}{M}$ and offline CL always employs the maximum $\alpha=\frac{TaskSize}{M}$). Our results reveal a complex relationship between the effect of $\alpha$ and loss function, data stream size, and memory budget size. Although proposition 1 suggests $\alpha_{min}$ is the optimal choice as long as $disc_L(P-,P+)>0$, we reckon that in practice, which $\alpha$ leads to the best performance likely depends on specific problems, algorithms, training regime and hyperparameter tuning may be needed. Nevertheless, the largest $\alpha$ value (i.e., the default choice of offline CL) appears to be far from optimal in many cases. Therefore, we believe a missed opportunity in the field of offline CL  is to experiment with $\alpha$ instead of always using the largest $\alpha$.
> >
> >
> > The importance of the storage allocation mechanism has likely remained under the radar due to standard practices in the field of CL: 1) focusing only on exemplar memory and overlooking the storage cost for task data, 2) treating online and offline CL as two separate research areas, and 3) usually applying a single iteration in the online setting. This has motivated our experiments with aligned settings for memory and computation.
> >
> > We hope this has clarified our motivation and finding of the aligned memory study. We will revise the introduction section accordingly to highlight this. Please let us know if there are any further concerns on this matter.

---

> ### Comment · Reviewer_QLLJ · 2023-11-17
> **Providing straws in the wind for the new experiment presented**
>
> Hi, thank you for engaging constructively despite my negative review. Hope these experiments and my review clarifies why online CL outperforming offline CL would be a tough sell. I find the result provided by the authors quite interesting. I am quickly replying to see if I can provide straws in the wind. I shall post my response to the broader rebuttal soon.
>
> - In scenarios with limited computation, computational constraints implicitly constrain memory (i.e. despite having access to more samples, an algorithm cannot really effectively utilize them). While this is highlighted in previous works exploring computational constraints in continual learning [1], I haven't seen an explicit analysis of this effect.
> - Hence, I think that the the experiment between (online-no-align) and (online-align-memory) is quite nice as it  in my view can help better understand the limits imposed by computation on memory use in CL methods more explicitly, by allowing access to a larger $C_{batch}$ compared to $C_{task}$.
> - In contrast, (online-align-compute) and (offline) result is better understood, the relative uselessness of $C_{batch}$ samples in contrast to memory $M_{online}$ was pointed out by GDumb [2]. In my understanding, they further removed even $C_{task}$ and only studied training on $M_{online}$ vs $M_{online}+C_{batch}$ and realized that the performance does not degrade very much. Although they did not perform such an clean, ablated experiment.
>
> I maybe wrong, please feel free to correct this if so.
>
> [1] Computationally Budgeted Continual Learning: What Does Matter?, CVPR23
>
> [2] GDumb: A Simple Approach that Questions Our Progress in Continual Learning, ECCV20

---

> ### Comment · Reviewer_QLLJ · 2023-11-17
> **Response to rebuttal**
>
> > However, we strongly disagree that this means experiments with aligned memory are not worth performing. 1) aligned memory allows systematic comparison of the usefulness of recent samples and past samples
>
> My objection is the conclusions drawn from the experiments with aligned memory. I did not say that any experiments are not worth performing -- my job is to simply evaluate the claims made based on the experiments. My claim still is that comparing aligned and offline memory isn't fair.
>
> > aligning memory budget provides a vital lens for studying efficient storage distribution constraints and representation mechanisms for CL. More specifically, 1) aligned memory allows systematic comparison of the usefulness of recent samples and past samples, and 2) it reveals the effect of the storage allocation mechanism for balancing samples from past and present on the performance of continual learning.
>
> I specifically request the authors to elaborate why they think aligned Online CL vs offline CL is a fair comparison. I'm afraid these  statements are too vague for me to agree or disagree, I would prefer a more concrete claim which I can either support or disagree with and give reasons why.
>
> > Does this statement always hold, regardless of data stream size, memory size, or CL algorithm (ER, SCR, iCaRL,DER etc)? And if so, why?
>
> The statement was intended to simply highlight the differences in distributional constraints of C and M. The distributional constraint remain the same regardless of the algorithm applied.
>
> I would add this effect of C being markedly less useful compared to M is quantified in the experiments provided by the authors in the rebuttals. Seeing the (offline) vs  (online-align-compute) comparison --  5000 $C_{batch}$ samples contribute to less than 1.5% of the total accuracy while 2000 $M_{online}$ contributes 30-38% accuracy, when computation is equalized and not implicitly constraining the memory itself!
>
> I still think the additional experiments support my hypothesis and list why, but am open to changing my mind. Currently as it stands the authors acknowledged my concerns in W1 but did not address it. Once this issue overarching all experiments is clarified, I can  discuss specific issues about the storage allocation policies etc.

---

> > ### Author Response · Authors · 2023-11-19
> > **Author reply to Reviewer QLLJ**
> >
> > We appreciate the reviewer raising this important discussion point. It allows us to further clarify our rationale for comparing continual learning algorithms under an aligned memory budget.
> >
> > ## why we argue that online and offline CL should be compared under an aligned memory budget?
> >
> > In our problem setting, we aim to learn from a potentially infinite data stream D(t) with a batch size of B, under a limited total memory space (e.g. in embedded devices).  _The historical exemplar samples, new task samples, and even the models all exist within this memory-constrained operational space_! However, current practices often only consider the storage cost of exemplar samples when comparing algorithms. A recent study [1] takes into account the combined storage cost of the model and exemplars, as some CL algorithms necessitate additional space to retain historical models and some do not. Nevertheless, we believe there is still an overlooked aspect concerning the storage cost of the new task samples. We argue that incorporating the new task samples within the memory allocation offers a more comprehensive comparison of CL algorithms under real-world memory constraints.
> >
> > Zhou, Da-Wei, et al. "A Model or 603 Exemplars: Towards Memory-Efficient Class-Incremental Learning." The Eleventh International Conference on Learning Representations. 2022.
> >
> > I hope this explanation clarifies the reasoning behind aligning the total memory budget. Please inform us if you have any other concerns or questions regarding this issue.

---

> ### Comment · Reviewer_QLLJ · 2023-11-22
> **Not Convinced: Reason Below**
>
> **Question**: Given say a total memory of n examples (say n=7000) in a class-incremental setup-- what performs better? Online continual learning or offline continual learning?
>
> **Core Difference**: The core difference is between sample size arriving, and not algorithms. Online continual learning algorithms gets access sequentially to 64 new samples of the new classes while Offline continual learning algorithms can potentially access  all the 5000 samples of the new classes at once if they wish to.
>
> The paper restricts offline continual learning to store all $C_{batch}$ samples as a part of $n$, consuming 5000 out of 7000 samples on the last set of classes, while allowing online continual learning to freely store $~7000$ samples across past data.
>
> **Ideal Sampling**: The ideal sampling (given within-class are i.i.d.) is to divide the 7000 sized buffer equally among past classes (achieved by reservoir sampling and class-balanced sampling both as no class imbalance)-- making sampling for online CL better than offline CL, hence resulting in better experimental results.
>
> *But why is this unfair?*
>
> **Offline CL can sample the same too!**: As I noted in (informal claim) in W1, offline continual learning algorithms, given this constraint, can simply choose to delete the vast amount of current samples instead of past samples with the same reservoir sampling or class-balanced sampling-- thereby performing equal to or better than any corresponding online continual learning algorithm.
>
> However, I raise my score to 3 as I understand the paper argues memory-constrained operational space should include all exemplars. However, my argument is that given this new constraint, offline CL algorithms should (and can) simply not store the vast amounts of new data.
>
> Why the difference in the alignment of past storage and current storage between works?
> 1. Privacy of past data-- This is the core argument used for differentiating them, by which works argue $C_{batch}$ and $M_{offline/online}$ have different constraints.
> 2. The rebuttal makes a cost argument to equalize the memory-- I do not understand why restricting storage of 100-250MB on hard disk is the bottleneck, while the algorithms presented require 2-10GB of GPU VRAM to run.

---

> ### Author Response · Authors · 2023-11-23
> **Author reply to Reviewer QLLJ: core difference between online and offline CL**
>
> We really appreciate the reviewer actively engaging in the discussion. However, we believe there is a misunderstanding of our problem setting.
>
> **1.Problem setting: data stream with batch size of B**
>
> >Core Difference: The core difference is between sample size arriving, and not algorithms. Online continual learning algorithms gets access sequentially to 64 new samples of the new classes while Offline continual learning algorithms can potentially access all the 5000 samples of the new classes at once if they wish to.
>
> To clarify, following [1] we assume online and offline CL are both dealing with the same problem setting (hence we compare them), i.e.
>  general continual learning:"_an infinite stream of training data where at each time step, the system receives a (number of) new sample(s) drawn non i.i.d from a current distribution D_t that could itself experience sudden or gradual changes._" [1]
>
> As argued in [1][2]  offline CL is a relaxation of this general CL setting with the additional knowledge of task boundaries to perform multiple passes in task data. ( "_Knowledge of the task boundaries is required (i.e. when tasks switch) and allows for multiple passes over large batches of training data. Hence, it resembles a relaxation of the desired continual learning system that is more likely to be encountered in practice_"[1])
>
> As shown in definitions 1 and 2 in our paper, we assume that task size T is the sequence of data with a static distribution in a data stream and batch size B is the sample size arriving in the data streams (B<<T).  Under this problem setting, we compare two ways of continual model update:
> 1) update the model as long as each incoming batch data comes: model update frequency = batch size B
> 2) wait until the task switch to collect a large batch of data (with the same distribution) before updating the model: model update frequency = task size T
>
> We find that 1) leads to better performance given aligned memory and iterations. The Rebuttal table below shows a summary of CL algorithms following these two ways. Thus, we believe achieving a deeper understanding of the efficiency of these two is important and is the focus of this work.
>
> If the batch size is equal to task size, B=T(data stream arrives in large batches), then online and offline CL are the same thing. However, we believe there are some real-world scenarios the sample size of arrival is not equal to static distribution data size (B<<T).
>
>
> [1]De Lange, Matthias, et al. "A continual learning survey: Defying forgetting in classification tasks." IEEE transactions on pattern analysis and machine intelligence 44.7 (2021): 3366-3385.
> [2] Buzzega, Pietro, et al. "Dark experience for general continual learning: a strong, simple baseline." Advances in neural information processing systems 33 (2020): 15920-15930.
>
>
>
> **2. store all task data**
>
> >The paper restricts offline continual learning to store all samples as a part of , consuming 5000 out of 7000 samples on the last set of classes, while allowing online continual learning to freely store samples across past data
>
> Saving the entire task data is the common practice of offline CL, not a novel restriction of this work. Based on our knowledge, all the offline CL algorithm in the table below uses the entire task data.
>
> We hope our results (aligning the computation only, aligning memory & computation together) pose questions to this current practice of the offline CL field: do you need to store all the task data when learning continual learning from a data stream.
>
> Rebuttal Table 1.
> |            | Algorithms                                                                            |
> |------------|-----------|
> | 1) online CL  | ER, ER-ACE, MIR, ASER, GSS, SCR, RAR, COPE, OCM,CBA                                             |
> | 2) offline CL |  iCaRL, BIC, IL2M, LUCIR, EEIL, RWalk, DER, MEMO, FOSTER, PODNET, FOSTER, WA, COIL, CO2L, DER++|
>
>
> **3. Data privacy**
>
> >Privacy of past data-- This is the core argument used for differentiating them
>
> We disagree the consideration of privacy differentiates task data or past data. The privacy of the current task data perhaps should also be considered. As offline CL needs to store the task data which may raise some privacy and data safety issue.
>
> **4.Storage cost of samples**
>
> >The rebuttal makes a cost argument to equalize the memory-- I do not understand why restricting storage of 100-250MB on hard disk is the bottleneck, while the algorithms presented require 2-10GB of GPU VRAM to run.
>
> We agree that on the current CL benchmark, the data samples consume less space than the model. The cost of the model has been discussed in other work. This work targets the current practice of focusing on the storage budget of exemplar samples. Instead, we argue that the storage of new samples should be considered together with the exemplar samples.

---

> ### Comment · Reviewer_QLLJ · 2023-11-23
> **Requesting Reply to other Weaknesses**
>
> > Saving the entire task data is the common practice of offline CL, not a novel restriction of this work. Based on our knowledge, all the offline CL algorithm in the table below uses the entire task data.
>
> I believe this point has been discussed quite in depth. The crux of disagreement between us remains that authors state that offline continual learning algorithms should not be allowed to adapt to the change in the problem constraint introduced here, while I think they should be allowed. Specifically, the claim that they *cannot* outperform online continual learning is a wrong claim in my view, because the paper intentionally did not adapt their sampling strategy to new aligned memory constraint.
>
> As a minor point, the DER paper itself states how to use their algorithm in an online fashion in appendix F.3 (also in Soutif-Cormerais etal noted in above rebuttal), iCARL and EWC++ (RWak) has been used in many online learning comparisons including Mai et. al. also noted in the above rebuttal. The dichotomy between algorithms is not as black and white as the authors claim, as one can use either of the online/offline algorithms in the other setting.
>
> Can the authors address the weakness 2? I still think Proof of Theorem 1 is incomplete, and would like the authors to address this before rebuttal deadline.

---

> ### Author Response · Authors · 2023-11-23
> **Author Reply to Reviewer QLLJ**
>
> We appreciate the reviewer for their time in assessing this paper and engaging in this long discussion.
>
> **1.Adapting offline CL to maintain a class-balanced memory by deleting samples, is this Gdumb?**
>
> If we are understanding correctly, the reviewer suggests that in our comparison, offline CL should be adapted to maintain a class-balanced memory by deleting some of task samples. We are not sure if the reviewer are referring to Gdumb. Gdumb maintains a class-balanced memory.
>
> Generally, how to adapt offline CL algorithms to achieve ideal sampling, does this mean deleting samples without training? If so, whether deleting samples with training will have a negative impact on the performance. We believe the answer to these questions is not straightforward, deserves detailed investigation and is not the focus of this work.
>
> However, we present some relevant results in the revised paper (section 6) on what is the effect of removing samples without training. We compare zero short memory vs. small short memory. In zero short memory case (like Gdumb),  each incoming batch is directly stored in the long-term memory by reservoir sampling. This means some samples are never seen by the model. Compared to $M_{short}=Batch size$, zero short memory leads to _worse_ performance, perhaps due to some samples are never used for the training. More discussion on this topic can be found in our reply to Reviewer LdF8 and in the revised paper.
>
> Split-CiFAR100
> |                      | ER     |  SCR
> |----------------------|----------------| ---------------|
> | M_short: task size | 39.8 $\pm$  0.7 |  47.2 $\pm$  0.6 |
> | M_short: batch size| 52.7 $\pm$ 0.8 | 56.3 $\pm$  0.1|
> |  M_short: zero    | 50.3 $\pm$1.4  | 52.5 $\pm$ 0.4 |
>
> Mini-ImageNet
> |                      | ER     |  SCR
> |----------------------|----------------| ---------------|
> | M_short: task size | 31.9 $\pm$2.8 |  47.4 $\pm$ 0.4 |
> | M_short: batch size| 43.9 $\pm$3.4 | 51.3 $\pm$ 0.5|
> |  M_short: zero    | 42.5 $\pm$ 2.7 | 45.6 $\pm$ 0.1 |
>
>
>
>
> **2. DER++, icarl, ewc, Rwalk have been tested in online setting but achieve much worse performance than offline setting**
>
> In theory, all the offline CL techniques should be applicable to online settings with zero or slight modification(changing herding to reserivor sampling). However, when applied in online manner, the same algorithms often report worse results. Offline CL survey papers (Masana et al.) reported iCaRL 33.5% with a 2k exemplar memory. Online CL survey papers (Mai et al.) reported iCaRL 19.2% with a 5k exemplar memory. This phenomenon leads to the misbelief that offline CL techniques (especially with knowledge distillation design) may not achieve good performance without multiple epoch training (this concern is explicitly expressed in DER paper). We show this phenomenon is because of the usage of memory and compute resources, not the online setting (or single epoch constraint) itself.
>
>
> **3.generalization bound**
>
> We agree that the generalization bound may not be tight. However, it should be note that this is not because of the missing proof of this paper, but because theorem 8 in Mansour et al. (2009)  (Eq 8 in our paper) is not tight. Based on our knowledge, how to derive tight bounds for transfer learning and how to measure the quality of the bound is still a challenge due to the distribution difference in the source and target domain. Achieving equality in theorem 8 already requires h*(Q1) =h*(Q2) , which may not be achieved in practice. However, this upper bound is meaningful when h*(Q1) and h*(Q2) are close, and this is usually the assumption of transfer learning. In our work, Q1 and Q2 are the true distribution of the data stream and the _expected distribution_ of the memory data. In the rehearsal-based methods, the memory data includes both new and past samples, we assume h*(Q1) and h*(Q2) are also reasonably close. As verified in Fig 3, this generalization bound seems to provide meaningful insights into the influence of data stream size, task size and memory budget size and the prediction that online outperforms semi-online cases in Fig 2b.

---

### Official Review · Reviewer_xja9 · 2023-10-31

**Soundness:** 3 good
**Presentation:** 2 fair
**Contribution:** 3 good
**Rating:** 6
**Confidence:** 5

**Summary:**

Online continual learning is considered much more challenging than offline continual learning due to the combined challenges of catastrophic forgetting and underfitting due to a single pass. This paper presents a unified framework to combine online and offline continual learning. It demonstrates that online continual learning becomes more performant than offline CL when both use the same memory and compute. It proposes a long-short-term memory based framework to unify online and offline CL paradigms. It presents a theoretical study on the generalization bound on online and offline CL.

**Strengths:**

This paper proposes a unified framework combining online and offline CL to achieve the best of two worlds i.e., online CL improves stability and offline CL improves plasticity.

It studies stability-plasticity tradeoff in online and offline CL and the impact of sequence length, buffer size and task size on their performance.

It presents generalization bound analysis using a proposed unified framework and ties it with memory allocation policy. And it presents theoretical justifications for why online CL performs superior to offline CL in various specified settings.

The experimental results on various benchmark datasets demonstrate that online CL outperforms offline counterparts when they both use the same compute and memory.

**Weaknesses:**

Although combining online and offline CL is interesting, using the same memory and compute for online CL as used by offline CL makes both computationally expensive. Online CL uses many more iterations at each timestep to match offline CL’s compute overhead. Thus online CL becomes ill-suited for many real-world applications e.g., on-device learning where fast adaptation / speed is critical. Online CL’s performance gains are mostly attributable to higher compute (more SGD updates) and memory usage.

This paper defines online CL based on batch learning i.e., learner receives a batch of new data and makes multiple iterations where each iteration combines a new batch with a sampled batch of old data from the buffer. This becomes very similar to offline CL with larger batch size. A more realistic online CL can be defined such that an online learner learns new data sample-by-sample manner (instead of batch-by-batch) using a single training iteration that combines a new sample with a batch of old data. This is much faster than the former definition and suitable for real world applications where speed matters. Another difference is the number of times each new sample is seen by the learner. Unlike the latter definition, the former one allows the model to see each new sample multiple times which is not truly a single pass (core ingredient of online CL).

I am not convinced that single pass causes underfitting and online CL requires multiple iterations at each timestep. Underfitting originates from specific design choices for example training from scratch where online CL model is randomly initialized and struggles due to lack of an optimum initialization (pre-training on a subset of data). There are methods that do not exhibit underfitting. An example of such a method is REMIND [1] that performs a base initialization using a subset (10% of ImageNet). REMIND performs online continual learning in sample-by-sample manner at each timestep with one SGD step / iteration on a mini-batch consisting of one new data and several old data. REMIND achieves competitive accuracy to the offline variant.

Although this paper presents valuable insights and observations, the proposed framework focuses mainly on how to balance memory and compute between online and offline CL. It does not propose any specific mechanisms to mitigate catastrophic forgetting, enhancing stability and plasticity and underfitting issues (when present due to design choices). It lacks sufficient scientific contributions.

It is unclear how much each online and offline CL model forgets due to absence of offline upper bound (jointly trained on all data).

[1] Hayes et. al., “Remind Your Neural Network to Prevent Catastrophic Forgetting”, In ECCV, 2020.

[2] Yasir et al., “A Real-time Evaluation in Online Continual Learning: A New Hope”, In CVPR 2023.

**Questions:**

Why do we need a unified framework combining online and offline CL from real-world application perspectives?

What is the offline upper bound (jointly trained on all data) which shows how much online CL / offline CL model forgets?

How practical is the online CL when using the same compute and memory as offline CL? Can this online CL keep up with the speed of data stream as described in this work [2]?

How does the proposed unified framework perform when a learner receives and immediately learns a single data point at each timestep i.e., sample-by-sample manner?

---

> ### Author Response · Authors · 2023-11-22
> **Author reply to Reviewer xja9**
>
> Thank you for taking the time to review our paper and provide feedback. We address your concerns and questions point by point as follows:
>
> **W1. multiple iterations in online CL**
> >Online CL uses many more iterations at each timestep to match offline CL’s compute overhead. Thus online CL becomes ill-suited for many real-world applications e.g., on-device learning where fast adaptation / speed is critical. Online CL’s performance gains are mostly attributable to higher compute (more SGD updates) and memory usage.
>
> To clarify, even with much smaller compute overhead, online CL with 10 iterations (48.4% $\pm$ 0.5) already leads to better performance than offline CL with 100 epochs (40.1% $\pm$ 0.8) as shown in Fig 1a. We analyze the influence of iteration number on stability and plasticity. With iteration <10, increasing iteration leads to better stability and plasticity at the same time and the performance boost is very significant. With iteration >10, increasing interaction starts to hurt stability (more forgetting) and the benefit on the overall CL accuracy is limited. One implication of our finding is, instead of always using a single iteration, one can adjust the iteration number based on the speed-accuracy requirements of the specific application.
> In addition, it should be noted that even with multiple iterations online CL still supports fast adaptation in the sense it updates the model as long as it receives an incoming batch. In contrast, offline CL waited until the collection of a task dataset to update the model. In an interactive application, online CL can make the system more responsive to user feedback.
>
>
>
> **W2. Definition of online CL: batch by batch vs. sample by sample**
>
> Our problem setting is that the incoming batch in the data streams comes with a batch size of B. This setting is commonly used in continual learning research works [1][2]. Under this problem setting, we compare two ways of continual model update:
>
> 1) update the model as long as each incoming batch data comes: model update frequency = batch size B
> 2) wait and collect a large batch of data before updating the model: model update frequency = task size T
>
> We find that 1) leads to better performance given aligned memory and iterations. The table below shows a summary of CL algorithms following these two ways. Thus, we believe achieving a deeper understanding of the efficiency of these two is important and is the focus of this work. We agree that the sample-by-sample data stream is an interesting case, but is out of the scope of this work.
>
> |            | Algorithms                                                                            |
> |------------|-----------|
> | 1) online CL  | ER, ER-ACE, MIR, ASER, GSS, SCR, RAR, COPE, OCM,CBA                                             |
> | 2) offline CL |  iCaRL, BIC, IL2M, LUCIR, EEIL, RWalk, DER, MEMO, FOSTER, PODNET, FOSTER, WA, RMM, COIL, CO2L, DER++|
>
>
> [1] Zheda Mai, Ruiwen Li, Jihwan Jeong, David Quispe, Hyunwoo Kim, and Scott Sanner. Online continual learning in image classification:An empirical survey. Neurocomputing, 469:28–51,2022.
>
> [2] Soutif-Cormerais, Albin, et al. "A Comprehensive Empirical Evaluation on Online Continual Learning." Proceedings of the IEEE/CVF International Conference on Computer Vision. 2023.
>
> **W3. I am not convinced that single pass causes underfitting.**
>
> [3] discusses the use of multiple iterations to deal with the underfitting-overfitting dilemma of online CL (when trained from scratch). Whether or not single pass causes underfitting is not the topic of this work. We simply show that with the same memory, compute and  CL algorithm, updating the model in an online manner leads to better performance than offline.
>
> [3] Zhang, Yaqian, et al. "A simple but strong baseline for online continual learning: Repeated augmented rehearsal." Advances in Neural Information Processing Systems 35 (2022): 14771-14783.
>
> **W4. This work does not propose any specific mechanisms to mitigate catastrophic forgetting, enhancing stability and plasticity and underfitting issues (when present due to design choices). It lacks sufficient scientific contributions.**
>
> We did not propose new algorithms. Instead, we study a fundamental problem concerning different CL algorithms, the usefulness of past samples vs. new samples. We highlight an overlooked but important factor concerning CL performance: the storage allocation policy$\alpha=M_{short}/M$. Online and offline CL are just variants of the same learning algorithm that only differ in the setting of the storage allocation parameter (Online CL: $\alpha=B/M$ and offline CL $\alpha=TaskSize/M$).
> Tuning the storage allocation parameter $\alpha$ leads to a bigger performance boost (better stability-plasticty tradeoff) than the specific algorithm design (Table 1). The effect of this parameter on forgetting (i.e.-stability) is presented in Fig 1b and Fig 5b.

---

> > ### Comment · Reviewer_xja9 · 2023-11-22
> >
> > Thank you for your response. I've read through the other reviews and responses. While I'm not sure if it is ICLR worthy, I do think there is value in pointing out the need to fairly compare online and offline CL under the same memory/compute budgets, and that the analysis performed is valuable. I am raising my score correspondingly.

---

### Author Response · Authors · 2023-11-23
**Summary of Changes in Currently-Revised Version**

We thank all reviewers for taking the time in reviewing this paper and providing valuable comments. We are revisiting our paper based on the feedback and comments provided and have submitted a currently-updated version. Here we summarize changes of the currently-updated version as detailed below:

- Add results of aligning iteration only in Section 3.3 (see Fig 1);
- Add results of using zero short-term memory in Section 6 and discuss differences in training procedure and relation to Gdumb;
- Revise the introduction to explain the rationale for aligning memory budget and highlight the finding regarding storage allocation parameter;
- Revise the writing in Section 4 to clarify our contribution and findings;
- Change the title to better capture the essence of our work:
old title: "Bridging the gap between online and offline continual learning", new title: "Revisiting online and offline continual learning through aligned resources"

Please let us know if you have any extra comments and suggestions.

---

### Meta-Review · Area_Chair_RPeY · 2023-12-07

**Metareview:**

(a) Summarize the scientific claims and findings of the paper based on your own reading and characterizations from the reviewers.
- The paper questions whether online continual learning (CL) is actually more challenging than offline CL, given equivalent memory and computation
- The authors find empirically that online CL can outperform offline CL
- The authors also provide a theoretical analysis which demonstrates that online CL yields a lower-generalization bound


(b) What are the strengths of the paper?
- The paper questions what initially seems like evidence (superiority of offline CL over online CL)
- The paper provides a common formalization of online and offline learning

(c) What are the weaknesses of the paper? What might be missing in the submission?
- Even after a significant discussion with the reviewers, there remain misunderstandings between authors and reviewers. These span the validity of the theoretical analysis, the fairness of the online vs. offline comparisons.

**Justification For Why Not Higher Score:**

This work goes against the current understanding that offline CL is less challenging than online CL. Unfortunately, the authors have not been able to convince the current pool of reviewers.

The discussion initiated by reviewer QLLJ regarding the memory distributional assumptions made in the online vs. offline case requires a discussion in the paper. For now, I cannot precisely conclude whether online CL can generally be better than offline CL or if this is rendered possible by standard methods for populating memories.

Further, and again as a result of a discussion initiated by reviewer QLLJ, elements regarding the applicability of the theoretical analysis in this setting need to be clarified. For example, elements of the discussion should be provided in the manuscript.

I also found that some of the questions raised by reviewer LdF8 also merit to be clarified in the next version of the manuscript. Similarly, clarifying the practical interest of the online CL setting (as defined in the paper) would also be helpful (see specific comments from reviewer xja9).

**Justification For Why Not Lower Score:**

N/A

---

### Decision · Program_Chairs · 2024-01-16

Reject